# Detection and Prediction of Macrophage Activation Syndrome in Still’s Disease

**DOI:** 10.3390/jcm11010206

**Published:** 2021-12-31

**Authors:** Clément Javaux, Thomas El-Jammal, Pierre-Antoine Neau, Nicolas Fournier, Mathieu Gerfaud-Valentin, Laurent Perard, Marine Fouillet-Desjonqueres, Julie Le Scanff, Emmanuelle Vignot, Stéphane Durupt, Arnaud Hot, Alexandre Belot, Isabelle Durieu, Thomas Henry, Pascal Sève, Yvan Jamilloux

**Affiliations:** 1Department of Internal Medicine, Croix-Rousse University Hospital, Hospices Civils de Lyon, Université Claude Bernard-Lyon 1, 69004 Lyon, France; clement.javaux@chu-lyon.fr (C.J.); thomas_3901@hotmail.fr (T.E.-J.); pierre-antoine.neau@chu-lyon.fr (P.-A.N.); nicolas.fournier@chu-lyon.fr (N.F.); mathieu.gerfaud-valentin@chu-lyon.fr (M.G.-V.); pascal.seve@chu-lyon.fr (P.S.); 2Department of Internal Medicine, Saint Luc Saint Joseph Hospital, 69007 Lyon, France; lperard@ch-stjoseph-stluc-lyon.fr; 3Department of Pediatric Nephrology, Rheumatology, Dermatology, Mère-Enfant Hospital, Hospices Civils de Lyon, Université Claude Bernard-Lyon 1, 69500 Bron, France; marine.fouillet-desjonqueres@chu-lyon.fr (M.F.-D.); alexandre.belot@chu-lyon.fr (A.B.); 4Department of Internal Medicine, Villefranche-sur-Saône Hospital, 69400 Gleize, France; JLeScanff@lhopitalnordouest.fr; 5Department of Rheumatology, Edouard Herriot University Hospital, Hospices Civils de Lyon, Université Claude Bernard-Lyon 1, 69003 Lyon, France; emmanuelle.vignot@chu-lyon.fr; 6Department of Internal Medicine, Lyon-Sud University Hospital, Hospices Civils de Lyon, Université Claude Bernard-Lyon 1, 69310 Pierre-Benite, France; stephane.durupt@chu-lyon.fr (S.D.); isabelle.durieu@chu-lyon.fr (I.D.); 7Department of Internal Medicine, Edouard Herriot University Hospital, Hospices Civils de Lyon, Université Claude Bernard-Lyon 1, 69003 Lyon, France; arnaud.hot@chu-lyon.fr; 8CIRI (Centre International de Recherche en Infectiologie), Inserm U1111, CNRS, UMR5308, ENS de Lyon, Université Claude Bernard Lyon 1, 69007 Lyon, France; thomas.henry@inserm.fr; 9Lyon Immunopathology Federation (LIFE), 69000 Lyon, France; 10Research on Healthcare Performance (RESHAPE), INSERM U1290, Université Claude Bernard Lyon 1, 69003 Lyon, France

**Keywords:** adult-onset Still’s disease, macrophage activation syndrome, systemic-onset juvenile idiopathic arthritis, ferritin, autoinflammatory disease

## Abstract

Distinguishing between macrophage activation syndrome (MAS) and a simple flare of Still’s disease (SD) may be challenging. We sought to clarify the clinical features and outcome of MAS in SD and to explore predictive factors of MAS development. Demographic and clinical data, treatments, and outcomes were recorded in a cohort of 206 SD patients. SD patients with and without MAS were compared. To explore predictive factors for the development of MAS, patients were compared at the time of SD diagnosis. Twenty (9.7%) patients experienced MAS, which was inaugural in 12 cases. Patients with MAS were more likely to have hepatomegaly (OR, 3.71; 95% CI, 1.14–11.2; *p* = 0.03) and neurological symptoms (OR, 4.43; 95% CI, 1.08–15.3; *p* = 0.04) than patients without MAS. Cytopenias, abnormal liver tests, and coagulation disorders were significantly more frequent in patients with MAS; lactate dehydrogenase and serum ferritin levels were significantly higher. An optimized threshold of 3500 μg/L for serum ferritin yielded a sensitivity (Se) of 85% and a negative predictive value (NPV) of 97% for identifying patients with/without MAS. Survival analysis showed that a high ferritin level at the time of SD diagnosis was predictive of MAS development (*p* < 0.001). Specific factors, including neurological symptoms, cytopenias, elevated LDH, and coagulopathy, may contribute to the early detection of MAS. Extreme hyperferritinemia at the onset of SD is a prognostic factor for the development of MAS.

## 1. Introduction

Still’s disease (SD) is a rare, non-monogenic autoinflammatory disorder characterized by high fever, evanescent rash, polyarthritis, and elevated WBC count [1,2]. By definition, adult-onset Still’s disease (AOSD) occurs in patients over the age of 16, but the continuum with systemic-onset juvenile idiopathic arthritis (SJIA) is now widely accepted [3,4,5]. The annual incidence rate of AOSD is estimated to be between 0.16 and 0.4 per 100,000 individuals, with a prevalence ranging from 0.3 to 2.7 per 100,000 individuals, depending on the country [6,7,8,9]. The pathophysiology of SD remains elusive. Excessive activation of innate immunity triggers the production of proinflammatory cytokines, such as interleukin (IL)-1, -18, -6, and interferon (IFN)-γ, along with nonspecific inflammatory biomarkers, such as ferritin [10]. Interestingly, during SD flares ferritin often reaches extreme levels, which are associated with a decrease in the glycosylated fraction of ferritin. The combination of elevated ferritin with a decrease in its glycosylated fraction (i.e., <20%) has been shown to be specific to SD flares [11,12].

SD has a good prognosis, with an estimated specific mortality rate of 1–3% [2,13,14,15]. However, some patients experience severe complications, such as fulminant hepatitis, myocarditis, or macrophage activation syndrome (MAS). MAS is a reactive hemophagocytic lymphohistiocytosis, which specifically complicates rheumatic diseases, such as systemic lupus erythematosus or SD [16]. The main manifestations of MAS are high fever, hepatosplenomegaly, cytopenias, coagulopathy, extreme hyperferritinemia, and hemophagocytosis on bone marrow aspirates. The occurrence of MAS has been reported in 5–19.5% of patients with AOSD and in 10% of patients with SJIA [2,17,18,19,20,21]. MAS can occur at the time of diagnosis or be secondary (i.e., during the course of SD). Specific predictive or diagnostic factors are lacking to help the clinician. Indeed, although it is an important criterion, hemophagocytosis is not specific to MAS [22,23,24].

In addition, MAS and SD flares share many clinical and biological signs and symptoms, making the identification of MAS difficult [4,25,26,27,28]. While scores have been developed for MAS diagnosis in SJIA [29,30,31,32], distinguishing a genuine MAS from a “simple” SD flare remains challenging and is a major management issue. MAS is a life-threatening condition with a high mortality rate of 10–20% in this situation [25,33], and can induce serious complications arising from cytopenias, risk of progression to disseminated intravascular coagulation, thrombotic microangiopathy, or multiple organ dysfunction syndrome. These conditions require a more aggressive management. While corticosteroids alone (or in association with standard doses of anakinra) may be sufficient to treat an SD flare, the existence of MAS, especially if there is an organ involvement, requires intensified therapy (high dose anakinra, etoposide, cyclosporine) [34,35,36]. Management in intensive care rather than conventional care may also be necessary. For all these reasons, the early identification of MAS is an important issue in the management of SD. Similarly, the predictive factors for the development of MAS during the course of SD have been little studied, often limited by the number of patients [37,38,39,40,41].

In this multicenter study, we sought to determine the specific characteristics that distinguish MAS from an SD flare and to explore the predictive factors of the secondary occurrence of MAS.

## 2. Patients and Methods

### 2.1. Study Design and Data Collection

All patients followed for SJIA or AOSD in nine regional clinical centers, between January 2001 and March 2021, were included in the observational cohort (AURAL STILL study). Patients were excluded if data at the time of diagnosis were missing. For each patient, data were collected from medical records and biological software using an anonymous and standardized electronic Case Report Form (2016 Ennov Clinical, CSOnline v.7.5.720.1). Demographic data, medical history, clinical, biological, and imaging data at the time of diagnosis were collected. If available, histopathological and other related test results were collected. Medical management, treatments, side effects, and outcome were recorded chronologically. Antinuclear antibodies (ANA) were considered positive if their titer was ≥160. HScore and HLH-2004 score were retrospectively calculated [42]. All data were collected at the time of SD diagnosis and at the time of MAS diagnosis. For patients with inaugural MAS, these data were the same, while data were different in the case of late-onset MAS.

### 2.2. Definitions

The Yamaguchi or Fautrel criteria for AOSD [12,43] and ILAR or PReS criteria for SJIA [44,45] were sought for all patients. In order to analyze patients with SD in real life, and to be able to evaluate the performance of the diagnostic criteria sets, the diagnoses of AOSD and SJIA were retained according to the final diagnosis of the clinician managing the patient (i.e., compatible clinico-biological presentation, disease course, treatments, and management). Diagnoses were also reviewed and confirmed by two study investigators (CJ, YJ). Monocyclic course was defined as a single systemic episode occurring for ≤1 year, with persistent complete remission for ≥1 year after cessation of any specific treatment. Polycyclic course was considered if patients had at least two episodes separated by ≥3 months of complete remission. Chronic course was defined as persistent active disease with predominant joint symptoms for ≥1 year [46]. MAS identification in the SJIA and SD groups was based on the final opinion of the referring clinician in charge of the patient and, therefore, when MAS diagnosis was specified in the medical record. MAS diagnoses were also retrospectively confirmed by two investigators (CJ, YJ) by compatible clinico-biological presentation, disease course, treatments, and management. We preferred such an assessment to reflect real life, keeping in mind the validated scores. HScore and HLH-2004 score were retrospectively calculated to evaluate their performance for the diagnosis of MAS [29,42]. MAS was categorized as follows: (i) inaugural MAS, if it was concomitant with the diagnosis of SD, or (ii) late-onset MAS, when it occurred after a period of disease remission. Patients were considered immunocompromised (i) if they had ongoing treatment with steroids (>10 mg per day during >2 weeks) or immunosuppressive drugs; or (ii) within 6 months after discontinuation of chemotherapy for malignancy; or (iii) after solid organ transplantation or hematopoietic stem cell transplant; or (iv) if they were HIV infected with uncontrolled viral load or AIDS; or (v) if they were hyposplenic or asplenic. Remission was defined as a clinical recovery of ≥3 months, with or without ongoing treatment. A flare was defined as the occurrence of systemic symptoms while the patient was in remission and requiring either resumption or intensification of treatment. Recovery was defined as inactive disease, without any treatment, after ≥1 year of follow-up. Death was categorized as disease-related or not.

### 2.3. Statistical Analysis

SD patients with and without MAS were compared, respectively, at the time of MAS onset and at the time of SD diagnosis to look for factors that could distinguish MAS from a simple SD flare (i.e., without MAS). This analysis was also dichotomized between SJIA (i.e., SD onset before the age of 16) and AOSD (onset after 16) to look for differences in more homogeneous groups. In a subsequent analysis, to look for predictive factors of secondary MAS, we excluded patients who had an inaugural MAS from the analysis, and compared the initial data of patients with late-onset MAS to the remaining patients without MAS. Descriptive analyses are presented as medians (interquartile range, IQR) for non-normally distributed continuous variables and as frequencies and percentages for categorical variables. For the percentage calculation of each variable, the number of missing values was excluded from the denominator (NB: parameters with ≥20% of missing data were excluded from the analysis). Nonparametric statistical methods were used to compare the study continuous variables by Mann–Whitney U-test, and the chi-square test or Fisher’s exact test as appropriate for categorical variables. A *p*-value of ≤0.05 was considered significant. A receiver operating characteristic (ROC) validation analysis was carried out by obtaining the area under the ROC curve to evaluate biological data and score in MAS prediction. The best cutoff was determined to be the one with the highest Youden’s index. Kaplan–Meyer curve analyses representing the probability of survival without MAS over time were compared using the maximally selected rank statistics from the maxstat and the survminer packages. All analyses were performed using the statistical software RStudio, v1 3.1093 (R foundation for Statistical Computing, Vienna, Austria).

### 2.4. Ethics

Ethical approval has been obtained from the institutional review board and the French CNIL (#19_348). This study was registered on ClinicalTrials.gov (accessed on 29 November 2021) under the ID #NCT05055882. All patients were sent written information about the study. In accordance with French legislation on non-interventional retrospective studies, no written informed consent was required for inclusion.

## 3. Results

### 3.1. Description of the Study Population

A total of 206 patients with SD were included, of whom 86 (42%) were male. The median age at the time of SD diagnosis was 27 (IQR, 11–43) years. A total of 78 (38%) patients had a diagnosis of SJIA. ILAR criteria were met in 43% of cases and PReS criteria in 53% of cases. Of the 128 patients with AOSD, 60% met the Yamaguchi criteria and 65% met the Fautrel criteria. Winter was the season with the highest frequency of SD onset (33%), while spring was the season with the lowest frequency of onset (19%). The median time from symptom onset to diagnosis of SD was 39 (IQR, 22–123) days. Epidemiological data, as well as clinical and laboratory findings, are shown in Table 1.

Fever and joint involvement were the most common symptoms. Arthritis was present in 45% (*n* = 77/171) of cases. The rash was typical (i.e., transient nonpruritic salmon-colored macules) in 13% (*n* = 18/135) of cases and was concomitant with fever peaks in 47% (*n* = 63/135) of cases. In total, 21% (*n* = 38/189) of patients had heart involvement, mostly pericarditis (*n* = 31/189, 16%), complicated by tamponade in five cases. Autoimmunity was present in 28% (*n* = 45/163), with antinuclear antibodies and rheumatoid factor being positive in 20% (*n* = 32/160) and 10% (*n* = 11/114) of cases, respectively. The disease course was mainly systemic, with 36% and 35% of monocyclic and polycyclic patterns, respectively (Table 1). More than half of the patients had at least one relapse after the diagnosis of SD (*n* = 109/189, 58%). Steroids were the first-line treatment used in 85% (*n* = 171/202) of patients, then IL-1-blockers (*n* = 85/202, 42%). Patients who did not receive corticosteroids often received a combination of other treatments: 11 patients with monocyclic form (NSAIDs, *n* = 10; anti-IL-1, *n* = 3; IVIG, *n* = 1), 11 with polycyclic form (NSAIDs, *n* = 5; anti-IL-1, *n* = 6; antiTNF, *n* = 2, IVIg, *n* = 1), 8 patients with chronic form (NSAIDs, *n* = 7; MTX, *n* = 2; anti-IL-1, *n* = 4; anti-IL-6, *n* = 2). At the end of the follow-up, 38% of patients (*n* = 72/192) were considered cured, while 2.7% (*n* = 5/185) of patients had died.

### 3.2. Specific Biological Parameters Are Useful for Distinguishing between MAS and SD Flare

We then compared SD patients with and without MAS at the time of MAS occurrence to look for factors that discriminate MAS from an SD flare. MAS complicated SD in 20 patients (9.7%). Twelve patients had inaugural MAS (median time, 30 days), while the remaining eight had late-onset MAS (median time, 600 days). MAS complicated 11 AOSD and 9 SJIA, with no preponderant seasonality. Compared to SD patients without MAS, those who developed MAS were more frequently immunocompromised (35 vs. 2.7%, *p* < 0.001). The only clinical features that were significantly different between SD patients with and without MAS were a higher frequency of hepatomegaly (32% vs. 11%, *p* = 0.025) and neurological symptoms (20% vs. 5%, *p* = 0.034; mostly headaches and epilepsy) in patients with MAS (Table 2).

Patients in the MAS group had significantly lower leukocyte counts (*p* < 0.001) and platelet counts (*p* < 0.001). They had higher ferritin levels (median, 13,444 μg/L (IQR, 4370–26,369), *p* < 0.001) but there was no difference in ferritin glycosylated fraction. Patients with MAS were more likely to have abnormal liver function tests, and their lactate dehydrogenase (LDH) levels were significantly higher (*p* < 0.001), as well as the median triglyceride levels (*p* = 0.001). They were more likely to have coagulopathy and hypofibrinogenemia (*p* = 0.009 and *p* < 0.001, respectively). Patients with MAS had no autoimmunity (*p* = 0.004). As expected, hemophagocytosis was more frequent in patients with MAS (*p* < 0.001, Table 3).

While no bacterial triggers were identified in the MAS group, a viral replication was observed in seven cases (35%) with a majority of EBV replication (*n* = 4). Overall, the HLH-2004 and the HScore were significantly higher in patients with MAS (*p* < 0.001).

There was no difference in steroid use, but interleukin blockers were more frequently used in patients with MAS (70 vs. 29%, *p* = 0.057; Table 2). In the MAS group, four patients (20%) required a transfer to an intensive care unit but did not develop multi-organ failure. A higher proportion of patients from the MAS group underwent an SD relapse during the follow-up (85 vs. 54%, *p* = 0.017), and a polycyclic course was more frequently observed in this group (55%). Finally, none of the patients from the MAS group had died at the end of the study period.

Although a continuum between AOSD and SJIA is admitted, we analyzed each condition separately to gain potential in-group homogeneity (Appendix A). There was a higher frequency of MAS in SJIA patients as compared to AOSD (11.5 vs. 8.5%, *p* = 0.48). However, this difference was not significant and the small size of the cohort precludes formal conclusions. Patients with SJIA + MAS were more frequently immunocompromised than other SJIA patients (*p* < 0.001). They had less joint involvement (*p* = 0.037) and a significantly lower prothrombin time (PT, *p* = 0.028). A greater proportion of SJIA + MAS patients received anti-IL-1 therapy (*p* = 0.033), intravenous immunoglobulins (IVIg; *p* = 0.018), and immunosuppressive drugs (*p* = 0.003), mostly ciclosporin. In 89% of cases, MAS occurrence was associated with a polycyclic course of SJIA. For AOSD patients, we found no additional differences in clinical and biological features. Yet, a greater proportion of AOSD + MAS received IVIg (*p* = 0.007) or anti-IL-6 therapy (*p* = 0.006).

### 3.3. The Detection of MAS Is Improved with Optimized Thresholds of Ferritin, LDH, and Fibrinogen

To better define the parameters that distinguish MAS from SD flares, we calculated the areas under the curves (AUC), the best thresholds, and associated sensitivities (Se) and specificities (Sp) for the most significant ones (Table 4).

The Youden’s index reached a maximum with a threshold of 3500 μg/L for serum ferritin, yielding a Se of 85%, a Sp of 62%, and a negative predictive value (NPV) of 97% for identifying patients with/without MAS. Similarly, an LDH level of 459 U/L yielded a Se of 90%, a Sp of 72%, and an NPV of 98%, and a fibrinogenemia of 4.39 g/L yielded an NPV of 94%. The best threshold for the HScore was 135, yielding a Se of 90%, a Sp of 85%, and an NPV of 99%. In our series, this cutoff value had a higher performance than the previously reported value of 169 [42].

### 3.4. Predictive Factors of MAS Development

The median time to MAS onset was 30 (IQR, 10–86) days. The survival analysis assessed the risk of MAS development by comparing SD patients according to their baseline ferritin levels. This analysis showed that patients with a ferritin >10,448 μg/L at the time of SD diagnosis had a significantly higher risk of developing MAS during the course of the disease (*p* < 0.001, Figure 1).

Finally, to explore the predictive factors of late-onset MAS, we compared SD patient characteristics at the time of SD onset, but excluding patients with inaugural MAS (Appendix A). Except a younger median age at the time of SD diagnosis (12 vs. 29 years, *p* = 0.012) and a lower median PT (*p* = 0.004), none of the characteristics present at the time of SD diagnosis were predictive of secondary MAS development. The proportion of patients with initial lung or heart involvement was greater in the group with late-onset MAS, but this difference was not statistically significant.

## 4. Discussion

MAS is a severe condition that can lead to multiple organ dysfunction syndrome, especially if not recognized early. The overlap between the symptoms of MAS and SD flares makes the identification of MAS challenging. In this large multicenter study, we combined data from adults and children with SD to maximize the chance of identifying discriminating factors. This combination was possible because previous studies have shown that these conditions share the same pathophysiology, clinico-biological features, disease course, and outcome [3,4,5,47]. Compared to the literature, the epidemiological and clinical characteristics of our patients were similar as those previously described, notably a slight female predominance and no obvious seasonality [2,48,49,50,51,52,53,54,55]. Due to the selection criteria (i.e., diagnosis based on the clinician’s opinion), which were chosen to reflect real life, we found that the classification criteria were met in only 60% of adults and 50% of children. In addition, we found autoimmunity in about 30% of patients, which is higher than previously described [2,13,49,52,56].

In our series, overt MAS occurred in 9.7% of cases, a frequency similar to those described previously [41,48]. Hepatomegaly and neurological symptoms were the only clinical features significantly over-represented in the SD + MAS group. In two previous cohort studies, SD + MAS patients had more frequent hepatomegaly, as well as splenomegaly and lymphadenopathy [37,40]. In addition, another study reported more frequent abdominal pain, without specifying its origin [38].

Neurological involvement has been reported in 10–73% of cases in reactive hemophagocytic lymphohistiocytosis (reHLH). Disorders such as encephalopathy, meningitis, epilepsy, or posterior reversible encephalopathy syndrome have been described [57,58,59]. In addition, Zhao et al. have observed a higher frequency of MAS in patients with AOSD and neurological symptoms [60]. Therefore, the occurrence of neurological involvement should prompt the clinician to suspect MAS in the context of an SD flare.

With regard to biological parameters, our results also confirm the need for close monitoring of leukocyte and platelet counts and fibrinogen levels. Indeed, while these biomarkers are usually increased during SD flare (due to biological inflammation), it is classic to suspect MAS when their levels are paradoxically normal or decreased. As expected, hemophagocytosis on bone marrow aspirates was more frequent in patients with MAS. However, it was also observed in 5% of SD patients without MAS. Such a finding supports a common pathophysiological spectrum encompassing MAS and SD [28,61]. Indeed, several authors have reported occult MAS in patients with SD, suggesting MAS may be integral to the pathogenesis of SD [17,62,63].

To facilitate the identification of MAS, we sought to determine the optimal thresholds for the most specific biomarkers. We found that, although this is a usual finding during SD flares, elevated ferritin levels were significantly higher during MAS [64]. A ferritin level <3500 μg/L had an NPV of 97% to rule out MAS. While previous publications have reported a good performance of low ferritin glycosylated fraction (i.e., <20%) for the diagnosis of SD [11], other authors have shown that this biomarker (i.e., <25%) can serve as a good marker for the diagnosis of reHLH, secondary to infection or hematological malignancy [65,66]. This biomarker also seems to be useful in other situations, such as in the worsening of COVID-19 [67], but our results indicate that it may not be useful to discriminate MAS from an SD flare (best cutoff, 21%; Se, 100%; Sp, 34%).

Conversely, we found significant differences in the HScore [42], which was higher in SD patients with MAS. We found an optimized diagnostic performance for a threshold of 135. This cutoff is slightly different from the one reported previously (i.e., 169) [42], but the HScore was developed for reHLH associated with infectious diseases and hematologic malignancies and may deserve to be optimized in the specific context of SD.

LDH has been repeatedly reported as a useful biomarker for the diagnosis of MAS, notably in SJIA [68,69], but it is not a component of the HScore. We found an optimized threshold of 459 U/L for the diagnosis of MAS (with a Se of 90%), suggesting that its addition should improve the performance of previously published diagnostic scores.

Like some previous studies, we showed that MAS is associated with the polycyclic course of SD and with a higher proportion of relapsing patients [38,70]. These results therefore suggest that the occurrence of MAS during the course of the disease, whether inaugural or not, could be a prognostic factor for a longer disease course, with a higher risk of relapse, requiring prolonged follow-up and good patient information.

Studies addressing the risk of late-onset MAS are scarce [38] and most have analyzed patients with inaugural and delayed MAS together [37,39,40,41]. We therefore attempted to identify prognostic factors and found that a higher ferritin at the time of SD diagnosis was associated with a higher risk of developing MAS. Interestingly, in the survival analysis, the threshold that best discriminated patients who will develop MAS indicated that these patients had extreme hyperferritinemia since the onset of SD. It is thus tempting to hypothesize that these patients would have a kind of hyper-reactivity, compatible with the concept of a “hyperferritinemic syndrome” [71], which would be principally responsible for the subsequent development of an MAS. To further explore the risk of developing a late-onset MAS, we excluded patients with inaugural MAS from the analysis and only found a younger age and a lower PT at SD onset. While it can be hypothesized that an earlier onset of SD seems to increase the probability of developing a serious complication or originates in an as-yet unidentified inherited defect, we have no explanation for the difference in PT and further studies will be needed to confirm this result. One hypothesis is that an occult underlying coagulopathy, only identified by this global and unspecific biomarker, could be present from the beginning of the disease and provide the basis for MAS in a later flare.

At odds with previous publications, we did not observe a significantly reduced survival rate in patients undergoing MAS [39,41], as all the SD + MAS patients were alive at the end of follow-up.

Although it has one of the largest sample sizes in the literature, our study has limitations. First, it has a retrospective design; there is a significant amount of missing data and the long recruitment period and changes in practices make the results more heterogeneous. Despite a large sample of SD patients, the number of cases with MAS remains limited, which may have prevented the identification of some prognostic factors. In the same way, statistical analysis should be interpreted with caution due to the small number of MAS patients. Finally, due to the absence of death in the MAS group, we were not able to perform a survival analysis.

## 5. Conclusions

MAS is a frequent complication and a challenging diagnosis in SD, mainly due to overlapping signs and symptoms. However, some specific factors, including cytopenias, elevated LDH, and coagulopathy, may help in its early detection. In contrast to the ferritin glycosylated fraction, the previously published HScore appears to be useful for the detection of MAS complicating SD. Its diagnostic performance in this particular context may be optimized. Extreme hyperferritinemia at the onset of SD is prognostic for the subsequent occurrence of MAS. However, larger studies are needed to clarify its exact value.

## Figures and Tables

**Figure 1 jcm-11-00206-f001:**
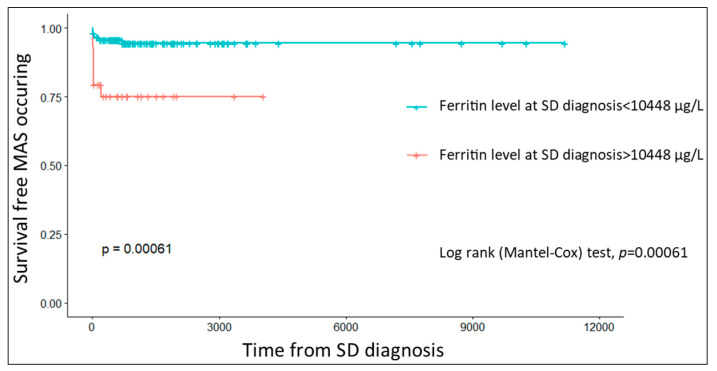
Kaplan–Meier curves showing the probability of MAS-free survival as a function of ferritin level at the time of SD diagnosis (MAS, macrophage activation syndrome; SD, Still’s disease).

**Table 1 jcm-11-00206-t001:** Patient characteristics at the time of diagnosis of Still’s disease.

Characteristics	*n* = 206
Epidemiology	
Age at diagnosis, median (IQR), y	27.0 (11.0–43.0)
Sex (male), No. (%)	86 (41.7%)
Caucasian origin, No. (%)	108/159 (67.9%)
AOSD, No. (%)	128 (62.1%)
SJIA, No. (%)	78 (37.9%)
Clinical	
Fever, No. (%)	183/196 (93.4%)
Constitutional symptoms, No. (%)	61/159 (38.4%)
Arthritis/arthralgia, No. (%)	176/199 (88.4%)
Skin rash, No. (%)	135/192 (70.3%)
Sore throat, No. (%)	92/183 (50.3%)
Lymphadenopathy, No. (%)	66/186 (35.5%)
Splenomegaly, No. (%)	22/157 (14.0%)
Hepatomegaly, No. (%)	20/153 (13.1%)
Lung involvement, No. (%)	37/189 (19.6%)
Heart involvement, No. (%)	38/189 (20.1%)
Digestive involvement, No. (%)	35/187 (8.7%)
Neurological involvement, No. (%)	14 (6.8%)
Laboratory	
WBCs, median (IQR), G/L	15.3 (11.4–19.7)
PMNs count, median (IQR), G/L	12.0 (8.2–15.9)
CRP, median (IQR), mg/L	160.0 (90.0–231.3)
Serum ferritin, median (IQR), μg/mL	2200 (489–8419)
Ferritin glycosylated fraction, median (IQR), %	14.0 (9.0–24.5)
Ferritin glycosylated fraction < 20%, No. (%)	63/97 (64.9%)
ANA, No. (%)	32/160 (20.0%)
Treatments	
Steroids, No. (%)	171/202 (84.7%)
IL-1-blockers, No. (%)	85/202 (42.1%)
IL-6-blockers, No. (%)	32/202 (15.8%)
Methotrexate, No. (%)	77/202 (38.1%)
Immunosupressive drugs, No. (%)	12/202 (5.9%)
IVIg, No. (%)	23/202 (11.4%)
Disease course	
Monocyclic (systemic), No. (%)	64/176 (36.4%)
Polycyclic (systemic), No. (%)	61/176 (34.7%)
Chronic (articular), No. (%)	51/176 (28.9%)

ANA, antinuclear antibodies; AOSD, adult-onset Still’s disease; CRP, C-reactive protein; IL, interleukin; IQR, interquartile range; IVIg, intravenous immunoglobulins; MAS, macrophage activation syndrome; PMNs, polymorphonuclear neutrophils; SJIA, systemic-onset juvenile idiopathic arthritis; WBCs, white blood cells.

**Table 2 jcm-11-00206-t002:** Comparison of clinical characteristics of SD patients with and without MAS.

Characteristics		With MAS (*n* = 20)	Without MAS (*n* = 186)	*p* Value
Epidemiology				
	Age at diagnosis, median (IQR), y	22.0 (12.0–27.0)	29.0 (11.0–43.8)	0.139
	Sex (male), No. (%)	7 (35.0%)	79 (42.5%)	0.685
	Caucasian origin, No. (%)	9/16 (56.2%)	99/143 (69.2%)	0.440
	Immunosuppression, No. (%)	7 (35.0%)	5/174 (2.7%)	<0.001
Classification criteria				
	ILAR, No. (%)	3/8 (37.5%)	24/55 (43.6%)	1.000
	PReS, No. (%)	3/7 (42.9%)	28/51 (54.9%)	0.694
	Fautrel, No. (%)	9/10 (90.0%)	74/118 (62.7%)	0.098
	Yamaguchi, No. (%)	9/10 (90.0%)	68/118 (57.6%)	0.050
Virus tests				
	Positive CMV PCR, No. (%)	0/13 (0.0%)	5/48 (10.4%)	0.574
	Positive EBV PCR, No. (%)	4/15 (26.7%)	10/51 (19.6%)	0.720
Clinical features				
	Fever, No. (%)	19/19 (100.0%)	165/178 (92.7%)	0.619
	Arthritis/arthralgia, No. (%)	14/18 (77.8%)	162/ 181 (89.5%)	0.137
	Skin rash, No. (%)	14/19 (73.7%))	120/174 (69.0%)	0.872
	Splenomegaly, No. (%)	2/19 (10.5%)	19/140 (13.6%)	1.000
	Hepatomegaly, No. (%)	6/19 (31.6%)	15/136 (11.0%)	0.025
	Lymphadenopathy, No. (%)	7/19 (36.8%)	57/168 (33.9%)	1.000
	Digestive involvement, No. (%)	4/19 (21.1%)	30/169 (17.8%)	0.754
	Lung involvement, No. (%)	3/19 (15.8%)	32/171 (18.7%)	1.000
	Heart involvement, No. (%)	6/19 (31.6%)	32/170 (18.8%)	0.225
	Neurological involvement, No. (%)	4 (20.0%)	10 (5.4%)	0.034
Treatments				
	Steroids, No. (%)	19 (95.0%)	152/182 (83.5%)	0.323
	IL-1-blockers, No. (%)	13 (65.0%)	72/182 (39.6%)	0.051
	IL-6-blockers, No. (%)	8 (40.0%)	24/182 (13.2%)	0.005
	Immunosupressive drugs, No. (%)	5 (25.0%)	7/182 (3.85%)	0.003
	IVIg, No. (%)	8 (40.0%)	15/182 (8.2%)	<0.001
Evolution				
	Recovery, No. (%)	6 (30.0%)	66/172 (38.4%)	0.305
	Relapse, No. (%)	17 (85.0%)	92/169 (54.4%)	0.017
	Death, No. (%)	0 (0.0%)	5/165 (3.0%)	1.000
Disease course				
	Monocyclic (systemic), No. (%)	6 (30.0%)	58/156 (37.2%)	0.530
	Polycyclic (systemic), No. (%)	11 (55.0%)	50/156 (32.1%)	0.042
	Chronic (articular), No. (%)	3 (15.0%)	48/156 (30.8%)	0.193

CMV, cytomegalovirus; EBV, Epstein–Barr virus; IL, interleukin; ILAR, International League of Associations for Rheumatology; IQR, interquartile range; IVIg, intravenous immunoglobulins; MAS, macrophage activation syndrome; PreS, Pediatric Rheumatology European Society; PCR, polymerase chain reaction.

**Table 3 jcm-11-00206-t003:** Comparison of biological characteristics of SD patients with and without MAS.

Biological Characteristics	With MAS *n* = 20	Without MAS *n* = 186	*p* Value
WBCs, median (IQR), G/L	6.0 (4.1–14.5)	15.4 (11.7–19.7)	<0.001
PMNs, median (IQR), G/L	11.2 (8.8–15.8)	12.0 (8.1–15.8)	0.919
Hemoglobin, mean (SD), g/L	107 (20.1)	113 (17.3)	0.222
Platelets, median (IQR), G/L	171 (130–210)	381 (273–509)	<0.001
Serum ferritin, median (IQR), μg/L	13,444 (4370–26,369)	2027 (460–7576)	<0.001
Ferritin glycosylated fraction, median (IQR), %	12.6 (3.00–16.0)	15.0 (10.0–25.0)	0.097
Ferritin glycosylated fraction < 20%, No. (%)	8/9 (88.9%)	56/89 (62.9%)	0.156
CRP, median (IQR), mg/L	142.0 (102.0–217.0)	162 (90.0–237.0)	0.907
ASAT, median (IQR), U/L	130 (84–242)	36 (24–59)	<0.001
ALAT, median (IQR), U/L	131 (71–216)	27 (15–62)	<0.001
GGT, median (IQR), U/L	132 (59–246)	50. (22–120)	0.024
LDH, median (IQR), U/L	678 (542–1236)	344 (242–484)	<0.001
PT, median (IQR), %	69.5 (60.8–82.2)	78.0 (69.0–93.2)	0.066
Fibrinogen, mean (SD), g/L	3.41 (2.11)	6.27 (2.06)	<0.001
Triglycerides, median (IQR), mmol/L	2.25 (1.87–3.08)	1.51 (1.14–2.07)	0.001
B2 microglobulin, median (IQR), mg/L	2.60 (2.40–2.76)	2.35 (2.16–2.95)	0.423
Autoimmunity, No. (%)	0/18 (0.0%)	45/145 (31.0%)	0.004
ANA, No. (%)	0/18 (0.0%)	32/142 (22.5%)	0.025
Hemophagocytosis, No. (%)	11/14 (78.6%)	4/79 (5.1%)	<0.001

ALAT, alanine–aminotransferase; ASAT, aspartate–aminotransferase; ANA, antinuclear antibodies; CI, confidence interval; CPK, creatinine phosphokinase; CRP, C-reactive protein; SD, standard deviation; GGT, gamma-glutamyltransferase; IQR, interquartile range; LDH, lactate dehydrogenase; MAS, macrophage activation syndrome; PMNs, polymorphonuclear neutrophils; PT, prothrombin time; WBCs, white blood cells.

**Table 4 jcm-11-00206-t004:** Diagnostic values of laboratory characteristics relevant to SD-associated MAS.

	Threshold	Sensitivity (%)	Specificity (%)	PPV (%)	NPV (%)
HScore > 169	>169	65.0	96.2	65.0	96.2
HScore > 135	>135	90.0	85.0	39.0	98.8
Ferritin (μg/L)	>3500	85.0	61.7	23.9	96.6
Ferritin glycosylated fraction (%)	<20	88.9	36.0	12.3	97.0
	<21	100.0	33.7	13.2	100.0
	<25	100.0	24.7	11.8	100.0
LDH (U/L)	>459	89.5	72.3	35.4	97.5
Platelet count (/mm^3^)	<230	80.0	86.2	38.5	92.0
Fibrinogen (g/L)	<4.39	72.2	80.9	57.0	94.0
Triglycerides (mmol/L)	>1.95	75.0	70.4	61.5	91.9
ASAT (U/L)	>84	80.0	83.3	41.0	96.6
ALAT (U/L)	>76	75.0	79.4	34.8	95.6

ALAT, alanine–aminotransferase; ASAT, aspartate–aminotransferase; LDH, lactate dehydrogenase; MAS, macrophage activation syndrome; NPV, negative predictive value; PPV, positive predictive value.

## Data Availability

Additional data may be obtained on reasonable request from the corresponding author.

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
