# Peer review of "Detection and Prediction of Macrophage Activation Syndrome in Still’s Disease"

_jcm, 2021, doi:10.3390/jcm11010206_

Round 1

Reviewer 1 Report

Thank you so much for the opportunity of reviewing the manuscript.

The main question, how authors identified MAS, because the groups consists of adult sJIA and SD, and for sJIA authors may apply criteria of A.Ravelii 2016, but how you identified MAS in SD. There are now similar criteria, as I know, for SD, which were validated the similar manner.

Please provide clear explanation about inclusion in MAS subgroup. Also please provide information about analysis in MAS group. What was a time point for anaylis collection? 

Author Response

The authors thank Reviewer 1 for his/her interest in our study. The following is a point-by-point response to the comments.

Major points:

  1. The main question, how authors identified MAS, because the groups consists of adult sJIA and SD, and for sJIA authors may apply criteria of A.Ravelii 2016, but how you identified MAS in SD. There are now similar criteria, as I know, for SD, which were validated the similar manner.

MAS identification in the SJIA and SD groups was based on the final opinion of the referring clinician in charge of the patient and therefore when the diagnosis of MAS was specified in the medical record. Diagnoses were also reviewed and confirmed by two study investigators (i.e. compatible clinico-biological presentation, disease course, treatments, and management). There are numerous HLH/MAS scores but most of them share common features (fever, cytopenias, organomegaly, liver dysfunction, coagulopathy with decreased fibrinogen or hypertriglyceridemia). When available, hemophagocytosis features on bone marrow aspirates/biopsy or organ biopsy were systematically taken into account. For the diagnosis of AOSD-associated MAS, we have preferred such an assessment to reflect real life, keeping in mind the validated scores (such as Ravelli’s one, Hscore, or HLH-2004). HScore (for cancer-associated HLH) and HLH-2004 score (for primary HLH) were retrospectively assessed to evaluate their corelation with the diagnosis of MAS. Debaugnies et al. 2016 evaluated the HScore and HLH-2004 score in a population of adult and pediatric MAS but not specifically in Still’s disease. Further studies are definitely required to validate SD-associated MAS. We have added: All data were collected at the time of SD diagnosis […] in the case of late-onset MAS.” (Page 3, lines 101-104).

  1. Please provide clear explanation about inclusion in MAS subgroup. Also please provide information about analysis in MAS group. What was a time point for analysis collection?

Data were collected at the time of SD diagnosis and at the time of MAS diagnosis. For patients with inaugural MAS, these data were the same while data were different in the case of late-onset (secondary) MAS (i.e. MAS occurred after an initial period of remission). Therefore, we compared data from patients without MAS at the time of SD diagnosis with those of patients at the time of MAS. This comparison was performed to identify the characteristics that could distinguish a simple SD flare (i.e without MAS) from a MAS.

In a subsequent analysis, we excluded the patients who had an inaugural MAS (i.e. at admission) from the analysis, and compared the initial data of patients with late-onset MAS to the remaining patients without MAS. This was the only way to look for predictive factors of secondary MAS occurrence (i.e. same method as Ruscitti et al. J Rheumatol, 2018). We have changed the text:MAS identification in the SJIA and SD groups was based […] to evaluate their performance for the diagnosis of MAS”. (Page 3, lines 118-123).

Reviewer 2 Report

Dear Authors, congratulations on your Manuscript 'Detection and prediction of Macrophage activation syndrome in Still's Disease: a cohort study of 206 cases.

I do have some questions, concerns and comments:

  1. The title is misleading if retrospective is not included. I would suggest to only use the first part of the title "Detection and prediction of Macrophage activation syndrome in Still's Disease'.
  2. We all know a prospective study would have been the ideal design however this is hard to do in this relatively rare disease
  3. In the introduction you tick all the boxes (disease of interest, problem of interest, gap in knowledge, aims) however I had to read twice to find the gap of Knowledge. Why is it so important to be able to distinguish MAS from a flare of disease, I would suggest to emphasize this a little better in a separate paragraph between the problem of interest MAS and your aims.
  4. Results: I assume the patients who did not receive steroids or any of the biologicals were monophasic patients. I would suggest to describe this group in one or two sentences after the first table, as this is the group, likely at lowest risk to develop a flare or MAS
  5. Results 3.2, line 183; it is noted the patients were often immunocompromised, does this mean you have done an immune work-up and there T-B cell repertoire was abnormal, or do you mean they were more likely to be treated with more immunomodulating medications? Please clarify this in the text.
  6. The numbers of MAS patients are minimal, so please take caution when reporting statistics on this group and how to interpret them. This should also be mentioned in the Discussion when you discuss the limitations in the last paragraph
  7. MAS seems more frequent in sJIA than AOSD, it would be good to comment on this and if this could have influenced the relative small numbers. 78 sJIA patient compared to 128 AOSD

Author Response

The authors thank Reviewer 2 for his/her comments. The following is a point-by-point response to the comments.

Major points:

  1. The title is misleading if retrospective is not included. I would suggest to only use the first part of the title "Detection and prediction of Macrophage activation syndrome in Still's Disease”.

We have modified the title accordingly.

  1. We all know a prospective study would have been the ideal design however this is hard to do in this relatively rare disease.

This is indeed an important point.

  1. In the introduction you tick all the boxes (disease of interest, problem of interest, gap in knowledge, aims) however I had to read twice to find the gap of Knowledge. Why is it so important to be able to distinguish MAS from a flare of disease, I would suggest to emphasize this a little better in a separate paragraph between the problem of interest MAS and your aims.

Distinguishing between a simple SD flare and genuine MAS, whether inaugural or complicating the course of the disease, is a major management issue, with therapeutic and prognostic implications. Although it is likely to accompany a significant number of flares, MAS is a complication of SD. Indeed, MAS is a life-threatening condition (with an estimated all-cause mortality of 25%, i.e. much higher than the overall mortality of SD, which is estimated to be between 0-3%). Immediately, complications arising from cytopenias, the risk of progression to DIVC (1-3% risk depending on the series), thrombotic microangiopathy, or multi-visceral failure require a more aggressive management. While corticosteroids alone (or in association with standard doses of anakinra) may be sufficient to treat a SD flare, the existence of MAS, especially if there is an organ involvement, requires intensified therapy (high dose anakinra, etoposide, ciclosporin). Management in intensive care rather than conventional care may also be necessary. The prognosis of MAS is conditioned by its early diagnosis and management. In sum, as in sepsis, MAS should not be considered as an accompanying condition but as an aggravating one, requiring specific management. For all these reasons, the early identification (or prediction) of MAS is an important issue in the management of SD. We have added a separate paragraph: “In addition, MAS and SD flare share many clinical and biological signs […] is an important issue in the management of SD”. (Page 2, lines 71-83).

  1. Results: I assume the patients who did not receive steroids or any of the biologicals were monophasic patients. I would suggest to describe this group in one or two sentences after the first table, as this is the group, likely at lowest risk to develop a flare or MAS.

While intuitively, patients who did not receive corticosteroids (or biotherapies) might seem to be those who had monocyclic systemic forms, the results are much more mixed (among those who did not receive corticosteroids: 11 monocyclic forms, 11 polycyclic forms and 8 chronic forms). Patients with monocyclic and polycyclic forms may have received unreferenced treatments (e.g. NSAIDs), or directly immunosuppressants, biotherapies or IVIG. In fact, patients who did not receive CS often received combinations of other treatments. We have detailed this : “Patients who did not received corticosteroids often received a combination of other treatments: 11 patients with monocyclic form (NSAIDs, n=10; anti-IL-1, n=3; IVIG, n=1), 11 with polycyclic form (NSAIDs, n=5; anti-IL-1, n=6; antiTNF, n=2, IVIg, n=1), 8 patients with chronic form (NSAIDs, n=7; MTX, n=2; anti-IL-1, n=4; anti-IL-6, n=2)” (page 5, lines 192-196).

  1. Results 3.2, line 183; it is noted the patients were often immunocompromised, does this mean you have done an immune work-up and there T-B cell repertoire was abnormal, or do you mean they were more likely to be treated with more immunomodulating medications? Please clarify this in the text.

Since the study was retrospective an immune work-up (except if it was available) was not performed. The definition of « immunocompromised patients » is given in the Methods : « Patients were considered immunocompromised (i) if they had ongoing treatment with steroids (> 10 mg per day during > 2 weeks), immunosuppressive drugs; or (ii) within 6 months after discontinuation of chemotherapy for malignancy; or (iii) after solid organ transplantation or hematopoietic stem cell transplant (iv) if they were HIV infected; with uncontrolled viral load or AIDS or (v) if they were hyposplenic or asplenic ». (Page 3, lines 125-129).

  1. The numbers of MAS patients are minimal, so please take caution when reporting statistics on this group and how to interpret them. This should also be mentioned in the Discussion when you discuss the limitations in the last paragraph.

We have added this limitation: “In the same way, statistical analysis should be interpreted with caution due to the small number of MAS patients.(Page 11, lines 370-372).

  1. MAS seems more frequent in sJIA than AOSD, it would be good to comment on this and if this could have influenced the relative small numbers. 78 sJIA patient compared to 128 AOSD.

Overall, 9/78 (11.5%) sJIA patients had MAS, while 11/128 (8.5%) AOSD patients had MAS. The difference is not statistically significant. Although there is a trend towards a higher frequency of MAS in sJIA it seems very complex to comment on this result, mostly because the sample size was small. Thus, we have added: There was a higher frequency of MAS in SJIA patients as compared to AOSD (11.5 vs 8.5%, p=0.48). However, this difference was not significant and the small size of the cohort precludes formal conclusions.” (Page 8, lines 240-243).

Round 2

Reviewer 1 Report

I am satisfied. The manuscript might be accepted